# Whole-Genome Sequence Analysis of *Flammulina filiformis* and Functional Validation of *Gad*, a Key Gene for γ-Aminobutyric Acid Synthesis

**DOI:** 10.3390/jof10120862

**Published:** 2024-12-12

**Authors:** Wenyun Li, Junjun Shang, Dapeng Bao, Jianing Wan, Chenli Zhou, Zhan Feng, Hewen Li, Youran Shao, Yingying Wu

**Affiliations:** 1College of Food Sciences & Technology, Shanghai Ocean University, Shanghai 201306, China; liwenyun1209@126.com (W.L.); shangjunjun@saas.sh.cn (J.S.); baodapeng@saas.sh.cn (D.B.); 2National Engineering Research Center of Edible Fungi, Key Laboratory of Applied Mycological Resources and Utilization of Ministry of Agriculture, Institute of Edible Fungi, Shanghai Academy of Agricultural Sciences, Shanghai 201403, China; wanjianing@163.com (J.W.); zhouchenli@saas.sh.cn (C.Z.); 3Jiangsu Chinagreen Biological Technology Co., Ltd., Siyang 223700, China; fz@chinagreenbio.com (Z.F.); hltk002@chinagreenbio.com (H.L.)

**Keywords:** monokaryon, *Hypsizigus marmoreus*, glutamate decarboxylase, molecular docking, heterologous expression, metabolic pathways

## Abstract

*Flammulina filiformis* is one of the widely produced edible fungi worldwide. It is rich in γ-aminobutyric acid (GABA), a non-protein amino acid with important physiological functions in humans. To investigate the functions of key genes in the GABA metabolic pathway of *F. filiformis*, we isolated the monokaryon *Fv-HL23-1* from the factory-cultivated *F. filiformis* strain *Fv-HL23* and then sequenced and assembled the genome using the PacBio Sequel and Illumina NovaSeq sequencing platforms. The results showed that the genome comprised 140 scaffolds with a total length of 40.96 Mb, a GC content of 49.62%, an N50 of 917,125 bp, and 14,256 protein-coding genes. Phylogenetic analysis based on the whole genome revealed a close evolutionary relationship of *Fv-HL23-1* with *Armillaria mellea*, *Lentinula edodes*, and *Schizophyllum commune*. A total of 589 carbohydrate-active enzymes were identified in the genome of *Fv-HL23-1*, suggesting its strong lignocellulose degradation ability, and 108 CYP450 gene family members were identified, suggesting important functions such as resistance to stress, secondary metabolite synthesis, and growth and development. The *F. filiformis* proteins glutamate decarboxylase 1 (Ff-GAD1) and glutamate decarboxylase 2 (Ff-GAD2), which may be responsible for GABA synthesis, were identified by protein alignment. Molecular docking analysis showed that Ff-GAD2 may have better catalytic activity than Ff-GAD1. To verify the function of *Ff-gad2*, its heterologous expression in the mycelia of the mononuclear *Hypsizigus marmoreus* was analyzed. Compared with wild type, the GABA content of mycelia was increased by 85.40–283.90%, the growth rate was increased by 9.39 ± 2.35%, and the fresh weight was increased by 18.44 ± 7.57%. Ff-GAD2 may play a catalytic role in GABA synthesis. In addition, the expression of the full-length *Ff-gad2* gene was increased by 7.96 ± 1.39 times compared with the exon expression level in *H. marmoreus* mycelia, suggesting that the intron may contribute to the heterologous expression of Ff-GAD2. Based on whole-genome sequencing, we analyzed the enzyme system related to the important life activities of *F*. *filiformis*, focusing on the function of Ff-GAD, a key enzyme in the GABA synthesis pathway. The results lay a foundation for elucidating the GABA metabolism pathway of edible fungi and developing targeted breeding strategies for GABA-producing edible fungi.

## 1. Introduction

*Flammulina filiformis* belongs to the genus *Flammulina*, family Physalacriaceae, and order Agaricales, and is a large fungus that can be used in both food and medicine. *F. filiformis* is one of the most produced edible fungi in industrial cultivation, accounting for more than 10% of the edible fungi worldwide [1]. China is the world’s largest producer of *F. filiformis*, with fresh production reaching 2.025 million tons by 2022 [2]. *F. filiformis* is not only delicious when fresh but also rich in bioactive compounds that are beneficial to the human body, including fungal immunomodulatory proteins, lectins, polysaccharides, flavonoids, terpenes, and polyphenols [3,4]. *F. filiformis* lowers blood pressure; exhibits anti-tumor, antioxidant, cholesterol-lowering, and anti-aging effects; and improves intestinal microbial metabolism [5,6]. Mining and analyzing key genes in the metabolic pathways of these bioactive compounds will help in the development and use of edible fungal resources. In recent years, the development of omics technology and the maturity of the genetic transformation systems for edible fungi have provided an effective means for this work [7].

Genomics studies the structure, function, and evolution of organismal genomes. It provides reliable gene sequence information for gene cloning, gene function research, and the analysis of metabolic pathways, signal transduction pathways, growth, and development, providing powerful tools and theoretical support for a deeper understanding of the genetic basis and evolutionary processes of life [8]. Park et al. [9] used genomic technology to demonstrate that *F. filiformis* can efficiently degrade lignin and produce alcohol dehydrogenase, which has potential applications in bioethanol production. *F. filiformis* cultivars can be divided into yellow and white varieties. However, they are prone to browning, leading to economic losses. Fu et al. [10] identified gene-encoding enzymes related to browning in the genome of the yellow strain Chuanjin NO. 3. These enzymes are involved in tyrosine metabolism and phenylalanine biosynthesis, forming a theoretical basis for subsequent postharvest preservation. Chen et al. [11] sequenced the genome of wild *F. filiformis* and predicted 13 terpenoid gene clusters but did not study the related biosynthetic pathways.

γ-aminobutyric acid (GABA) is a ubiquitous non-protein amino acid in plants, animals, bacteria, and fungi, and it plays a role in promoting growth and development and responding to stress in plants [12]. In animals, it contributes to lowering blood pressure, regulating blood glucose, relieving anxiety and insomnia, and regulating tumor immunity [13,14]. A GABA shunt coupled to the tricarboxylic acid cycle is an important GABA metabolic pathway in a variety of organisms, including fungi, in which glutamate decarboxylase (GAD) catalyzes the key reaction step [15]. α-ketoglutarate in the TCA cycle is catalyzed by glutamate dehydrogenase to produce L-glutamate (L-Glu). Under the action of GAD, the α-site of L-Glu undergoes an irreversible decarboxylation reaction to produce GABA. With the development of molecular biology technologies, *gad* genes have been cloned and expressed in a variety of animals, plants, and microorganisms, and the biochemical characteristics of their corresponding GAD proteins have been preliminarily explained [16,17]. However, research on GABA in edible fungi has mainly focused on content detection, and its synthetic mechanism has not been reported. Among edible fungi, the content of GABA is relatively rich in *F. filiformis*, reaching 229.7–1080 μg/g DW in the fruiting body [18,19,20,21], which can be used as an excellent source material for the study of GABA metabolic pathways and related functional genes in edible fungi.

To explore the whole-genome characteristics of *F. filiformis*, the screened commercial strain *Fv-HL23* with high GABA yield was used. The monokaryon was isolated, and the whole genome was sequenced using PacBio Sequel and Illumina NovaSeq. Moreover, evolutionary relationships among the genomes of 22 published common fungi were analyzed. The lignocellulose degradation ability of *F. filiformis* was analyzed based on the variety and quantity of the carbohydrate-active enzymes (CAZymes). Furthermore, in silico function prediction was performed for the *Ff-gad*-coded protein, a key gene for GABA synthesis identified within the *Fv-HL23-1* genome, and manipulated using an efficient genetic transformation technology system built by the research team. *Ff-gad* was expressed heterologously in the mycelia of the monokaryon body of *Hypsizigus marmoreus*, the GABA content of which was much lower than that of *F. filiformis*. Moreover, the Ff-GAD function was verified in vitro. The results demonstrate a feasible scheme for the analysis of the GABA metabolic pathway of *F. filiformis* and provide a scientific basis for the directed breeding of edible fungi rich in GABA and the development of related products.

## 2. Materials and Methods

### 2.1. Experimental Strains and Plasmids

The *F. filiformis* dikaryon strains *W1638*, *Fv-YH*, *Fv-RYJ*, *Fv-HTC*, *Fv-HL23*, *Fv-SY*, *Fv-GR*, *Fv-GF*, *Fv-FM*, *Fv-CYS*, *Fv-BY*, *J54-3*, *J5011*, *WH25*, *X3E*, *ENOKI-J*, *ENOKI-I*, *ENOKI-H*, *ENOKI-G*, *2345(Y)*, and *SCY1-2(Y)* and the monokaryon *H. marmoreus* strain *Finc-W247-F4* used in this experiment are preserved by the National Edible Fungi Germplasm Resource Bank (Shanghai), Ministry of Agriculture and Rural Affairs, P.R. China, and the source information of 22 strains of *F. filiformis* is shown in Table 1. The p018-empty vector and p019-HmGPD-uvGFP plasmids were deposited in our laboratory, and the specific plasmid maps are shown in Appendix A. *Agrobacterium tumefaciens* (EHA105) was purchased from Shanghai Weidi Biotechnology Co., Ltd., Shanghai, China, and *Escherichia coli* (DH5α) was purchased from Vazyme Biotech Co., Ltd., Nanjing, China.

### 2.2. Culture Medium

Complete yeast (CYM) liquid medium was prepared as follows: peptone (2 g), yeast powder (2 g), KH_2_PO_4_ (0.5 g), MgSO_4_·7H_2_O (0.5 g), glucose (20 g). Distilled water was added to 1 L, and the medium was sterilized at 121 °C for 20 min.

Potato dextrose agar (PDA) medium was prepared as follows: potato dextrose agar (39 g) was added to 1 L of distilled water and sterilized at 121 °C for 20 min.

Induction medium (IM) was prepared as follows: K-buffer (10 mL), M-N buffer (20 mL), 20% glucose (*w*/*v*; 10 mL), 0.01% FeSO_4_ (*w*/*v*; 10 mL), 20% (NH_4_)_2_SO_4_ (*w*/*v*; 2.5 mL), 1% CaCl_2_·2H_2_O (*w*/*v*; 1 mL), 50% glycerin (*w*/*v*; 10 mL). The pH was adjusted to 5.6, and distilled water was added to 1 L. After sterilization at 121 °C for 20 min, the temperature was reduced to 50 °C, and then 1 mol/L 2-(N-morpholine) ethosulfonic acid and 100 mmol/L acetosyringone were added.

Yeast malt glucose (YMG) solid medium was prepared as follows: yeast extract (4 g), malt extract (10 g), glucose (10 g), and agar (15 g), plus distilled water to 1 L and 20 min sterilization at 121 °C.

Yeast extract peptone dextrose (YPD) liquid medium was prepared as follows: yeast extract (10 g), soy peptone (20 g), and glucose (20 g), with distilled water to 1 L and 20 min sterilization at 121 °C.

Cultivation medium was prepared as follows: corn cob (33.79%), rice bran (31.07%), bran (9.32%), cotton seed shell (7.77%), soybean husk (5.44%), beer grains (3.88%), dried bean residue (3.5%), beet residue (3.11%), shell powder (1.55%), and light calcium carbonate (0.58%), with 20 min sterilization at 121 °C.

### 2.3. F. filiformis Fruiting Body Cultivation

Each bottle contained 970–990 g of cultivation material, and 35–40 mL of liquid strain was added. The strain was cultured at 15 °C for 20–23 d at 70% relative humidity. Then, the fungus was placed in a growth chamber at 14–15 °C for 8–10 d until the mushroom bud was full of material surface. Then, the mushroom bud was continuously inhibited by light and wind, the CO_2_ concentration was controlled at 7000–15,000 mg·L^−1^, the temperature was gradually cooled to 4–5 °C, and the fruiting body emerged from the mouth of the bottle. When the mushroom bud was 1 cm out of the bottle mouth, the coating was applied. After the coating, once the mushroom sprout grew a further 1–2 cm out of the bottle mouth, CO_2_ was maintained above 10,000 mg·L^−1^ to promote stalk elongation, and the temperature was controlled at 5–7 °C. When the length of *F. filiformis* reached harvesting height at 15–16 cm, samples were collected using the four-point sampling method, with three parallel samples in each group.

### 2.4. Determination of GABA Content in the Fruiting Body of F. filiformis

According to the four-point method, 150 g of the mature fresh *F. filiformis* fruiting body was accurately weighed, dried in an oven at 55 °C to a constant weight, and crushed into powder using a high-speed grinder (BJ-400T, Deqing Baijie, Huzhou, China). About 60 mg powder of the sample was weighed out and added to 1 mL solution (0.1 mol·L^−1^ hydrochloric acid to 10% trichloroacetic acid; 1:2 *v*/*v*); this was followed by adding 2–3 steel balls and homogenizing the sample using a grinding machine. Ultrasonic extraction was performed for 20 min, followed by centrifugation at 2000 r·min^−1^ for 20 min. Approximately 0.5 mL of supernatant was removed and held at 4 °C for 30 min; this was then centrifuged for 20 min, and 0.4 mL of the supernatant was collected. The GABA content in the samples was determined using an automatic amino acid analyzer (L-8900, Hitachi, Tokyo, Japan), with three parallel samples in each group.

### 2.5. F. filiformis Monokaryon Genome Sequencing and Analysis

#### 2.5.1. Clamp Connection Observation Under Fluorescence Microscope

A 6 mm diameter mycelial block of *Fv-HL23-1* isolated by the protoplast method [22] was inoculated on YMG medium. The mycelial block was placed on a sterile cover glass at 45° at the edge of the colony and incubated at 25 °C until 67% confluence. A 20 μL drop of DAPI staining solution (Beyotime Biotechnology, Shanghai, China) was placed in the center of the slide and incubated in the dark for 10 min. The excess staining solution was removed with lens cleaning paper. The isolated strains were then examined using an inverted fluorescence microscope (Axio observer 3, Carl Zeiss, Oberkochen, Germany) at 100× magnification to observe the presence or absence of clamp connections.

#### 2.5.2. Mycelium Collection

The strain *Fv-HL23-1* was inoculated on CYM liquid medium and incubated with shaking at 150 r·min^−1^ for 5 d at 23 °C. Under aseptic conditions, 50 g mycelium was collected, flash-frozen in liquid nitrogen, and then stored at –80 °C.

#### 2.5.3. Genome Sequencing and Assembly

The Illumina NovaSeq (paired-end, 2 × 150 bp, insert size, 400 bp) and PacBio Sequel sequencing (CLR Mode, insert size, 10 K) platforms from Personalbio Technology (Shanghai, China) were used to sequence the *Fv-HL23-1* genome.

Falcon’s CANU v2.1.1 software was used to assemble sequencing read data de novo to construct contigs and scaffolds [23]. Illumina sequencing reads were filtered using Fast QC v0.11.9 software. Illumina short reads were corrected using Pilon v1.18 software [24], and BUSCO v3.0.2 was used to assess the integrity of the genome assembly [25].

#### 2.5.4. Gene Prediction and Annotation

Three software programs, Augustus v3.03 [26], GlimmerHMM v3.0.1 [27], and GeneMark-ES v4.35 [28], were used to predict the de novo gene model of the genome; Augustus v3.03 prediction was also performed with 3720 single-copy genes from the *Agaricales* family in the BUSCO database. EvidenceModeler v2.0.0 software was used to integrate gene predictions from scratch [29]. RepeatModeler v1.0.4 [30] and RepeatMasker v4.0.5 [31] were used for de novo and homology annotations.

Functional annotation of predicted protein-coding sequences (CDSs) based on the NCBI Nr, Kyoto Encyclopedia of Genes and Genomes (KEGG), Gene Ontology (GO), Pfam-A, and Swiss-Prot databases was performed using KAAS v2.1 [32], InterPro v66.0 [33], and Diamond v0.9.10.111 software [34]. EggNOG-mapper v4.5 software was used to predict and annotate the functions of the CDSs [35].

#### 2.5.5. Gene Family and Species Tree Construction

Genomic data for GCA_021015755.1 (*Lentinula edodes*), GCA_003813185.1 (*Lentinus tigrinus*), GCA_000300575.1 (*Agaricus bisporus*), GCA_014466165.1 (*Pleurotus ostreatus*), GCA_000143565.1 (*Laccaria bicolor*), GCA_015179015.1 (*Boletus edulis*), GCA_030407055.1 (*Armillaria mellea*), GCA_022496215.1 (*Russula brevipes*), GCA_013053245.1 (*Agrocybe pediades*), GCA_000271645.1 (*Tremella mesenterica*), GCA_000271585.1 (*Trametes versicolor*), GCA_000143185.2 (*Schizophyllum commune*), GCA_000182895.1 (*Coprinopsis cinerea*), GCA_000827495.1 (*Hypholoma sublateritium*), GCA_000827485.1 (*Amanita muscaria*), GCA_003444635.2 (*Morchella importuna*), GCA_000225605.1 (*Cordyceps militaris*), GCA_012934285.1 (*Ophiocordyceps sinensis*), GCA_000151645.1 (*Tuber melanosporum*), GCA_003070745.1 (*Tuber borchii*), GCA_000146045.2 (*Saccharomyces cerevisiae*), and GCA_000167675.2 (*Trichoderma reesei*) were downloaded from the NCBI online database. The protein sequences from 16 species of Basidiomycota and 7 species of Ascomycota, including *Fv-HL23-1*, were screened for length, and those with fewer than 50 amino acids were removed. The contrast threshold was set to 1 × 10^−10^, and the contrast sequence length was set to 70% using OrthoFinder v2.5.4. Markov Clustering was used to cluster the gene families. Multi-sequence alignment of single-copy orthologous genes of each species was performed using MAFFT software version 7 (https://mafft.cbrc.jp/alignment/software/) (accessed on 20 March 2024), and a phylogenetic tree of the species was constructed using the ML algorithm in FastTree 2.0.0 software (http://www.microbesonline.org/fasttree/) (accessed on 28 March 2024) for visualization.

#### 2.5.6. Identification of CAZymes

The CAZymes of the above 23 fungal genomes were annotated using hmmscan v3.2.1 software [36], and ORF sequences with lengths of >80 amino acids were selected with the E-value threshold set to 1 × 10^−5^. If the sequence alignment length was less than 80 amino acids, the E-value cut-off was set to 1 × 10^−3^, and the amino acid sequence alignment length was more than 30% of the amino acid sequence in the database.

#### 2.5.7. Cytochrome P450 Gene Family Identification and Phylogenetic Tree Construction for *F. filiformis*

The P450 amino acid sequence of *Armillaria mellea DSM 3731* in the P450 protein sequence database was used as the search object, and the selection conditions were E-value ≤ 1 × 10^−5^ and Identity ≥40. Blastp (2.5.0+) software was used to predict the P450 gene in the *Fv-HL23-1* genome. The Conserved Domains Database (https://www.ncbi.nlm.nih.gov/Structure/cdd/wrpsb.cgi) (accessed on 20 November 2024) and STRING database (https://cn.string-db.org/) (accessed on 22 November 2024) were used to compare the conserved domains of amino acid sequences of candidate CYP450 gene families to identify the members of the *Fv-HL23-1* CYP450 gene family. The Muscle program in MEGA v11 software [37] was used for multiple sequence comparisons, and the phylogenetic tree was constructed using the adjacency method and the bootstrap test method for 1000 repetitions.

### 2.6. Mining and Bioinformatics Analysis of Key GABA Synthesis Gene Gad

GAD protein sequences were downloaded from the NCBI online database, and BLASTP alignment was performed using BioEdit v7.0.9.0. The NCBI Conserved Domains Database (https://www.ncbi.nlm.nih.gov/Structure/cdd/wrpsb.cgi) (accessed on 27 October 2024) was used to test the conservative domain structure, ExPASy Protparam to analyze the basic information of proteins (https://web.expasy.org/protparam/) (accessed on 29 October 2024), SOPMA to predict the secondary structure of proteins (https://npsa.lyon.inserm.fr/cgi-bin/npsa_automat.pl?page=/NPSA/npsa_sopma.html) (accessed on 3 November 2024), AlphaFold2 to predict the tertiary structure of proteins (https://github.com/sokrypton/ColabFold/blob/main/AlphaFold2.ipynb) (accessed on 7 November 2024), PyMOL v3.0 and AutoDock Tools v1.5.7 software to perform molecular docking analysis, MEGA v11 software to construct the phylogenetic tree using the neighbor-joining method [37], MEME Suite (https://meme-suite.org/meme/) (accessed on 14 November 2024) to analyze conserved motifs, and TBtools v2.056 software to perform visualization [38].

### 2.7. Construction of Heterologous Ff-gad2 Expression Plasmid in F. filiformis

Based on the previous research of our team, using the endogenous mutant gene *Pesdi1* of *Pleurotus eryngii* as a safety screening marker [39,40] and the endogenous gene *HmGPD* (glyceraldehyde-3-phosphate dehydrogenase, GPD) of *H. marmoreus* as a promoter, we replaced the *Hpa* I-*Xba* I fragment in the transformation plasmid p019-HmGPD-uvGFP with the *Ff-gad2* gene to construct heterologous expression plasmids [41].

The p019-HmGPD-uvGFP plasmid was digested with *Hpa* I and *Xba* I restriction endonucleases (New England Biolabs, Beijing, China) to recover the target linear fragment. Ff-gad2-gDNA-F and Ff-gad2-gDNA-R1/Ff-gad2-gDNA-R2 primers were used to amplify the *Ff-gad2-gDNA* gene from the genome of strain *Fv-HL23-1*. The linearized fragment was ligated to the *Ff-gad2-gDNA* gene to construct the pWY601 plasmid using the homologous recombinase of the ClonExpress Ultra One Step Cloning Kit (Vazyme, Nanjing, China). The predicted *Ff-gad2-cDNA* gene was synthesized and ligated to the linearized p019-HmGPD-uvGFP, producing the pWY603 plasmid. This process was performed by Sangon Biotech Co., Ltd. (Shanghai, China). The primer design is detailed in Table 2, and the plasmid maps and primer positions of pWY601 and pWY603 after construction are detailed in Appendix A.

### 2.8. Transformation and Validation of Ff-gad2 in H. marmoreus

The pWY601, pWY603, and p018-empty vector plasmids were transferred into *A. tumefaciens* EHA105 using the freeze–thaw method, and the arthroconidia of the monokaryon bodies of *H. marmoreus* were used as the acceptor material. Using the *A. tumefaciens*-mediated genetic transformation method established by BAO et al. [40], we transferred allogeneic expression plasmids of *Ff-gad2* into wild-type *Finc-W247-F4*. Then, 10^6^/mL arthroconidia of *H. marmoreus* were co-infected with the plasmid containing *A. tumefaciens* strain EHA105 in IM and cultured in a dark box at 28 °C for 2 d. The cells were eluted with 0.05% Tween-20 and cultured in YMG medium containing 2 mg/mL carboxin and 300 μg/mL cefotaxime for screening. After 10 d of incubation at 23 °C, single colonies were selected for secondary screening. Subsequently, the DNA of putative pWY601, pWY603, and p018-empty vector transformations cultured at 23 °C for 15 d was collected. According to the transformation plasmid design, Ff-gad2-gDNA-F/Ff-gad2-gDNA-R2, Ff-gad2-cDNA-F/Ff-gad2-cDNA-R, and Pesdi1-F/Pesdi1-R were amplified and verified by sequencing (Table 2).

### 2.9. Determination of T-DNA Integration Sites in H. marmoreus

Plate hyphae of pWY601 and pWY603 cultured at 23 °C for 15 d were collected, genomic DNA was extracted from the samples, and standard library construction was performed using Illumina’s TruSeq DNA PCR-free prep kit. Paired-end sequencing of the library was performed on the Illumina NovaSeq sequencing platform with 400 bp reads. Whole-genome re-sequencing was performed by Personalbio Technology. The sequences from RB (right border) to LB (left border) of the pWY601 and pWY603 plasmids were aligned to the genome of the transformants using Blastp in BioEdit v7.0.9.0 software, allowing for the identification of the precise T-DNA insertion locations in the genome.

### 2.10. Real-Time Fluorescence Quantitative PCR of Exogenous Ff-gad2 in H. marmoreus

The total RNA of the inverters was extracted using the FastPure Plant Total RNA Isolation Kit (Vazyme, Nanjing, China) according to the HiScript III RT SuperMix for qPCR (Vazyme, Nanjing, China). The transcriptional level of *Ff-gad2* was detected using real-time fluorescence quantitative PCR (7500, Thermo Fisher, Waltham, MA, USA). *Actin* from *H. marmoreus* was selected as the internal reference gene, and the transcription level of the pWY603-3 strain was set to 1.0. The relative transcript levels of other strains were expressed as fold changes relative to the reference transcript level of strain pWY603-3. The primers q-Ff-gad2-F/q-Ff-gad2-R and q-ACT1-F/q-ACT1-R were used to amplify *Ff-gad2* and *Actin*, respectively (Table 2). Taq Pro Universal SYBR qPCR Master Mix (Vazyme, Nanjing, China) was used to perform RT-qPCR experiments, and the relative gene expression was calculated using the 2^−ΔΔCT^ method, with three parallel samples in each group.

### 2.11. Determination of GABA Content of Transformations

The transformations were activated on PDA plates for 14 d, and 20 pieces of 8 mm^2^ culture blocks were inoculated in triangular YPD medium bottles with a liquid volume of 100 mL/250 mL. After 12 d of shaking at 23 °C and 150 r·min^−1^, the hyphae were collected, dried in an oven at 55 °C to constant weight, and crushed using a high-speed pulverization machine to obtain the sample powder. The GABA content was determined after treatment, with three parallel samples in each group.

### 2.12. Mycelial Growth Rate and Aerial Mycelial Fresh Weight Determination in Transformations

Mycelium growth rate was measured using the criss-crossing method [42]. After inoculation, a cross was made on the back of the PDA plate, with the culture block mass as the central point, and the tip was marked after the mycelium germinated. Once the mycelium covered two-thirds of the Petri dish, markings were made on the peripheral-most part of the mycelium, and the average mycelial growth rate was calculated as the distance between the two markings divided by the number of days of cultivation, with three parallel samples in each group.

The weight of fresh airborne mycelia was determined using the method described by Cao et al. [43]. Cellophane membrane was spread on PDA medium, and then the culture blocks were inoculated on the center of the membrane and cultured at 23 °C for 14 d. The airborne mycelium was collected, and its fresh weight was determined, with three parallel samples in each group.

### 2.13. Data Analysis

Three replicates were used for each group. SPSS software (version 2.0) was used for data significance analysis, and GraphPad Prism 9.5 software was used for plotting. Different lowercase letters indicate significant differences (*p* < 0.05).

## 3. Results and Analysis

### 3.1. Screening of F. filiformis Strains with High GABA Content and Isolation of Monocytic Strains

The GABA content in the fruiting bodies of *F. filiformis* strains cultured under uniform factory conditions was between 1500 and 4000 mg/kg DW, as measured using an amino acid analyzer (Figure 1A). Among them, *Fv-HL23* had the highest GABA content of 3915.68 ± 661.16 mg/kg DW, and *Fv-HL23* had better fruiting body agronomic traits (Figure 1B). Therefore, strain *Fv-HL23* was selected for further studies. Monocytic strains of *Fv-HL23* were isolated using a protoplast preparation method. Using fluorescence inverted microscope microscopy, we observed that the *Fv-HL23* mycelium was a dikaryotic strain with a clamp connection (Figure 1C). However, the isolated monokaryon *Fv-HL23-1* showed no clamp connections (Figure 1D). In addition, the growth rates of *Fv-HL23* and *Fv-HL23-1* were 3.50 ± 0.32 mm/d and 1.88 ± 0.26 mm/d, respectively, after being cultured at 25 °C for 18 d, which was consistent with the research results that binucleate strains grew faster than monokaryon strains [44,45,46].

### 3.2. Gene Assembly and Analysis of Monosomal Strain Fv-HL23-1

Genome sequencing of *Fv-HL23-1* was performed using the PacBio Sequel and Illumina NovaSeq sequencing platforms. A depth distribution map generated using GenomeScope showed a heterozygosity of 0.02%, further demonstrating that *Fv-HL23-1* was a monokaryotic strain (Appendix A). A summary of *F. filiformis* gene annotation is shown in Table 3, with a full genome size of 40.96 Mb, which was reassembled into 140 contigs with a GC content of 49.62%, N50 of 917,125 bp, N90 of 141,141 bp, and a maximum contig length of 2,467,745 bp (Figure 2). Compared with other fungi, the size of the *Fv-HL23-1* genome was comparable to that of *S. commune* (38.7 Mb), *L. tigrinus* (39.5 Mb), and *A. muscaria* (40.7 Mb). The genome was larger than those of *S. cerevisiae* (12.1 Mb), *T. mesenterica* (28.6 Mb), *A. bisporus* (30.2 Mb), *C. militaris* (32.3 Mb), *T. reesei* (33.4 Mb), *P. ostreatus* (34.9 Mb), and *C. cinerea* (36.2 Mb) but smaller than those of *T. versicolor* (44.8 Mb), *A. pediades* (45.1 Mb), *L. edodes* (45.6 Mb), *H. sublateritium* (48.0 Mb), *R. brevipes* (48.5 Mb), *M. importuna* (50.9 Mb), *L. bicolor* (64.9 Mb), *B. edulis* (66.5 Mb), *A. mellea* (70.9 Mb), *T. borchii* (97.2 Mb), *O. sinensis* (110.9 Mb), and *T. melanosporum* (124.9 Mb) (Appendix A). The small genomes of *S. cerevisiae* and *T. reesei* could be attributed to simplification to facilitate environmental adaptation during long evolutionary processes [47,48]. The genomes of *O. sinensis* and *T. melanosporum* are particularly large because of biological characteristics such as a complex life history, diverse environmental adaptability, parasiticity, and symbiosis. The existence of a large number of transposon repeat elements leads to significant genome expansion [49,50,51], suggesting that *F. filiformis* does not have the relevant biological characteristics of these species.

The genetic structure of most fungi is simple, and it is beneficial to improve structural estimation accuracy by comparing gene prediction models [52]. Based on both homologous and de novo annotations, repetitive sequences accounted for 1.36% of the total *Fv-HL23-1* genome, among which, the main types were LTR and DNA transposons, accounting for 1.05% and 0.14%, respectively (Table 4). A total of 14,256 genes were predicted in the *Fv-HL23-1* genome, with an average sequence length of 1649.9 bp. The total length of the CDSs was 19,949,251 bp, accounting for 55.56% of the entire genome. To predict the protein sequences, similarity analysis was performed on 14,256 non-redundant genes from 9 public databases. The P450 database yielded the most matches (13,953 genes/97.87%), followed by the NCBI NR (12,167 genes/85.35%), EggNOG (9638 genes/67.61%), Pfam (8351 genes/58.58%), SwissProt (6955 genes/48.79%), GO (13,953 genes/48.28%), KEGG (3814 genes/26.75%), PHI (2416 genes/16.95%), and TCDB (1530 genes/10.73%) databases. By annotating and classifying genes using different databases, we can better understand their functions and characteristics.

The whole-genome integrity of *Fv-HL23-1* was determined using BUSCO v3.0.2 to be 96.1%, which is greater than 90.0% [53]. This was similar to that reported for *Lyophyllum decastes* (95.7%) [54] and *Pleurotus giganteus* (94.2%) [55] and higher than that of *Hericium rajendrae* (91.6%) [56] and *Agaricus bitorquis* (90.8%) [57]. In summary, we obtained a high-quality *Fv-HL23-1* genome for further analyses.

### 3.3. Phylogenetic Analysis

Orthologous genes are products of speciation, and the description of orthologous relationships among species helps to reconstruct their evolutionary processes. In addition, homology analysis is the most accurate method for identifying similarities and differences between species, inferring functional genetic information from model organisms, and proposing functions for newly sequenced genomes. To predict the orthologs of *Fv-HL23-1* and 22 other fungal species, we generated Venn diagrams of gene families by clustering the proteomes using *Fv-HL23-1* and OrthoMCL v2.0.8 software (Appendix A). Proteome clustering showed that among the 14,256 genes predicted for *Fv-HL23-1*, 8081 were shared by multiple species, including 1486 single-copy homologous genes, which are presumed to play important roles in evolution and genetic variation in *F. filiformis*. In addition, 1794 core orthologous genes were found in the genomes of 23 fungal species and are presumed to be important for fungal life activities, such as adapting to environmental changes and maintaining normal physiology, growth, development, and genetic stability (Appendix A).

To study the evolutionary relationships of *Fv-HL23-1* based on the results of homologous gene cluster analysis, single-copy homologous genes were selected for multiple sequence alignment, and the ML algorithm in FastTree 2.0.0 was used to construct a phylogenetic tree. The values on the branches of the phylogenetic tree indicate branch reliability, where the closer to 100, the more reliable, and the bootstrap value is 100% of the highly supported internal branches, accurately reflecting the availability of data, as well as the evolutionary relationships of different fungal species. The 23 fungal species were divided into 4 groups. *Fv-HL23-1* belonged to the same group III as *A. mellea*, *L. edodes*, and *S. commune*, and the genetic distance between *Fv-HL23-1* and *A. mellea* was the smallest. In addition, *O. sinensis*, *C. militaris*, *T. reesei*, *M. importuna*, *T. melanosporum*, *T. borchii*, and *S. cerevisiae* all belonged to Ascomycota and were categorized into group I. However, *T. mesenterica*, *R. brevipes*, *L. tigrinus*, *T. versicolor*, *B. edulis*, and *P. ostreatus* and *A. bisporus*, *A. muscaria*, *C. cinerea*, *L. bicolor*, *A. pediades*, and *H. sublateritium* belonged to Basidiomycota and were classified into independent groups II and IV, respectively, indicating a certain conservation in their evolutionary relationships (Figure 3).

### 3.4. CAZyme Analysis

CAZymes are important enzymes involved in biomass hydrolysis and synthesis and in the degradation and utilization of organic matter in macrofungi [58]. A total of 589 CAZyme-encoding genes were annotated in the *Fv-HL23-1* genome. These genes included 86 encoding glycosyltransferases (GTs), 22 encoding polysaccharide lyases (PLs), 105 encoding carbohydrate esterases (CEs), 102 encoding auxiliary activities (AAs), 23 encoding a family of carbohydrate-binding domains (CBMs), and 251 encoding a family of glycoside hydrolases (GHs) (Figure 4A). To investigate the proteins involved in organic matter degradation by *Fv-HL23-1*, the major protein families of the six CAZymes in *Fv-HL23-1* were mapped. The GTs were mainly distributed in the GT1, GT2, GT8, GT32, and GT68 families; the PLs in the PL1, PL3, PL8, and PL14 families; the CEs in the CE1, CE4, CE10, CE12, and CE16 families; the AAs in the AA1, AA3, AA5, AA7, and AA9 families; the CBMs in the CBM1, CBM13, CBM19, CBM21, and CBM50 families; and the GHs in the GH5, GH13, GH16, GH18, and GH43 families. These proteins are speculated to play important roles in the degradation of lignin, hemicellulose, pectin, and starch and in the cleavage of polysaccharides (Figure 4B) [9,59,60]. Moreover, a comparative analysis of the 23 fungi regarding CAZymes revealed that *Fv-HL23-1* contained abundant CAZymes, ranking the fifth highest (589) below only *A. mellea* (735), *A. pediades* (659), *P. ostreatus* (607), and *L. tigrinus* (597) (Figure 4C). These results suggest that *Fv-HL23-1*, a low-temperature wood-decaying fungus, can use cellulose and hemicellulose as substrates for mycelial growth and has strong organic matter degradation activity. This is consistent with the fact that agricultural and forestry products rich in lignocellulose, such as sawdust from broadleaf trees, straw, and cotton seed shells, are often used as raw materials for *F. filiformis* cultivation. Overall, the 23 fungi were divided into two groups: Basidiomycota and Ascomycota. The number of annotated CAZymes for 16 Basidiomycota ranged from 212 to 735, including 60–92 GTs, 3–26 PLs, 29–136 CEs, 13–144 AAs, 16–64 CBMs, and 73–288 GHs. The number of CAZymes annotated for seven Ascomycota ranged from 153 to 445, with 58–90 GTs, 0–23 PLs, 13–66 CEs, 11–75 AAs, 5–18 CBMs, and 57–210 GHs. In general, there were more CAZymes in Basidiomycota than in Ascomycota, with comparable distributions of GTs and PLs but more CEs, AAs, CBMs, and GHs in Basidiomycota than in Ascomycota. These genes are speculated to play key roles in the biological and physiological properties of Basidiomycota fungi, including the degradation and utilization of organic matter, maintenance of cell wall integrity, elasticity and tensile strength, ion homeostasis, and responses to abiotic stresses [61,62,63].

### 3.5. Analysis of Cytochrome P450 in F. filiformis

Cytochrome P450 (CYP450) is a large enzyme superfamily that includes heme as a cofactor. It widely exists in animals, plants, bacteria, fungi, and other living organisms, acting as a monooxygenase. In eukaryotic cells, CYP450 is mainly distributed in the microsomes, endoplasmic reticulum, and mitochondrial inner membrane [64]. The development and maturity of genome sequencing technology in recent years have greatly promoted the identification, classification, and expression analysis of the fungal CYP450 gene family. The P450 gene family has been confirmed to reduce the toxicity of fungal xenochemicals [65], promote growth and development [66], promote the synthesis of secondary metabolites [67], and promote the interaction between fungi and plants to adapt to the environment [68].

A total of 108 CYP450 gene family members were identified in the *Fv-HL23-1* mononuclear genome via homologous alignment and detection of the CYP450 protein domain, which is consistent with the reported total number of p450 (107) genes in the mononuclear *F. filiformis Liu355* [11]. As seen in Appendix A, the lengths of the 108 CYP450 proteins range from 81 to 885 aa, and their molecular weights range from 9.16 to 100.32 kDa. To better understand the genetic relationships among members of the *Fv-HL23-1* CYP450 gene family, a phylogenetic tree was constructed based on the results of multiple sequence comparisons and was divided into four categories: A, B, C, and D. Groups A and B were closely related, while other the groups were distantly related. FfCYPs on different branches were suspected to perform different functions (Figure 5). In addition, based on KEGG data analysis, 64 out of 108 CYP450s were annotated to metabolic pathways. They were classified into 27 subfamilies: CYP1, CYP2, CYP3, CYP4, CYP6, CYP7, CYP8, CYP12, CYP17, CYP21, CYP27, CYP28, CYP46, CYP49, CYP51, CYP61, CYP67, CYP81, CYP82, CYP86, CYP94, CYP98, CYP313, CYP708, CYP735, CYPD, and CYPH (Table 5). According to the protein functions of these families, the 64 FfCYPs may be involved in important life processes in *F. filiformis* such as resistance to stress, synthesis of secondary metabolites, and growth and development.

### 3.6. Bioinformatics Analysis of Gad in F. filiformis and H. marmoreus

The GAD protein sequences of *L. edodes* (XP_046088738.1), *H. marmoreus* (RDB24023.1), *P. eryngii* (KAF9493791.1), *A. mellea* (KAK0197355), *Lactobacillus brevis* (AB258458.2) [69], and *Arabidopsis thaliana* (NP_197235.1) [17] were downloaded from Genbank. BioEdit v7.0.9.0 software was used with the blastp *Fv-HL23-1* genome to establish a local protein library, and the corresponding two proteins with the highest similarity were obtained as scaffold24.g57 and scaffold24.g63, which were submitted to the STRING database for verification and named Ff-GAD1 and Ff-GAD2, respectively. The protein scaffold4.g242 with the highest similarity in the Finc-W247-F4 genome was also aligned, submitted to the STRING database for verification, and named F4-Hm-GAD2.

The relative positions of exons and introns of the two GAD-coding genes in *Fv-HL23-1* were visualized, and Ff-gad1 had 16 exons and 15 introns, whereas Ff-gad2 had 17 exons and 16 introns; the similarity between the two genes was as high as 85.68% (Appendix A). Ff-GAD1 and Ff-GAD2 had instability indices lower than 40 and grand average hydropathicity lower than 0, suggesting that they are stable hydrophilic proteins with amino acid residues of 554 and 536 and molecular weights of 61.89 kDa and 59.60 kDa, respectively, with subcellular localization in mitochondria (Table 6). Prediction of the conserved protein sequence domains showed that both Ff-GADs had a specific conserved region, GadA, belonging to the pyridoxal phosphate-dependent aspartate aminotransferase superfamily (AAT-I superfamily), and contained a conserved pyridoxal 5′-phosphate (PLP) binding site in the middle (Appendix A). The secondary structure of a protein is based on a helix and a folded polypeptide chain so that the amino acids form a stable spatial structure. The secondary structure predicted by SOPMA showed that Ff-GAD1 contained 213 α-helices (38.45%), 74 extended strands (13.36%), 28 β-folds (5.05%), and 239 random coils (43.14%). Ff-GAD2 contained 208 α-helices (38.81%), 69 extended strands (12.87%), 29 β-folds (5.41%), and 230 random coils (42.91%). Some differences were identified in the secondary structures of the two Ff-GAD proteins, but they were mainly α-helix and random coil structures, which have high stability and folding performance, with specific biological functions achieved by the complex spatial structure (Appendix A). The arrangement of multiple secondary structural elements in 3D space forms a compact 3D structure called the tertiary structure of a protein molecule. AlphaFold2 was used to predict the tertiary spatial structure models of Ff-GAD1 and Ff-GAD2. The predicted local distance difference test yielded values of 85.9 and 88.3, respectively. The protein translational modification values were 0.868 and 0.882, respectively, indicating that both proteins were highly reliable. Both proteins contained a large number of α-helices and random coils, which was consistent with the secondary structure prediction. The results of the protein tertiary structure confidence map showed that the two Ff-GADs contained only one polypeptide chain, and their spatial structure was relatively stable (Appendix A).

The binding affinity represents the binding ability of the component to the target. The lower the binding ability, the more stable the binding of the ligand and receptor, and a value below –5.0 kcal/mol indicates better binding activity [70]. L-Glu (Compound CID: 33032) and PLP (Compound CID: 1051) were obtained through the PubChem database. The main active component was optimized using PyMOL software to remove water molecules and small molecular ligands, and hydrogenation and charge treatment were performed using AutoDock Tools. The key targets were the receptor and its corresponding active component as a ligand. After simulating the interaction of Ff-GAD proteins with natural substrate L-Glu and cofactor PLP, the binding affinities of Ff-GAD1 and Ff-GAD2 with L-Glu were −5.4 kcal/mol and −5.9 kcal/mol, respectively, and those with PLP were −5.5 kcal/mol and −6.2 kcal/mol, respectively. In addition, L-Glu formed hydrogen bonds with the Gln112, Thr137, Ala138, and Ser352 binding sites of Ff-GAD1 (Figure 6A). Hydrogen bonds were formed with the Tyr331, Phe333, Ser334, and Phe337 binding sites of Ff-GAD2 (Figure 6B). PLP formed hydrogen bonds with the Val183, Tyr233, and Lys552 binding sites of Ff-GAD1 (Figure 6C) and the Thr137, Ala138, Ser332, Ser334, and Asn336 binding sites of Ff-GAD2 (Figure 6D). Both L-Glu and PLP interacted with Ff-GAD through hydrogen bonding to form stable complexes.

A phylogenetic tree of the two GAD proteins from *Fv-HL23-1* was constructed with those from other edible and medicinal fungi, along with *A. thaliana*, *Levilactobacillus brevis*, *S. cerevisiae*, and *T. reesei*, resulting in seven groups (groups I–VII) (Figure 7A). Ff-GAD1 and Ff-GAD2 clustered in the same branch and GAD from *Cylindrobasidium torrendii*, *Hymenopellis radicata*, and *Armillaria* spp., forming group IV together with GAD from other genera. This suggests that these proteins are closely related. In our previous study, F4-Hm-GAD2 from the genome of Finc-W247-F4 was located in group V, suggesting it was distantly related to the two Ff-GAD proteins. GAD from different strains of *Mycena* spp., *Lentinula* spp., and *Armillaria* spp. were clustered in the same group. The macrofungi all clustered in a large branch formed by groups I–VI, and group VII contained the plant *A. thaliana*, the prokaryote *L*. *brevis*, the eukaryote *S. cerevisiae*, and the filamentous fungus *T. reesei*, suggesting that the clustering of GAD is consistent with the evolutionary relationships of the species.

Differences in the number of conserved motifs and the composition of domains may affect the function of the proteins encoded by their genes. As shown in Figure 7B, the conserved structural domains of Ff-GAD in *Fv-HL23-1* and the rest of the GAD proteins from the same source were mainly categorized into three major groups: the AAT-I, Glu-decarb-GAD (phosphopyridoxal-dependent glutamate decarboxylase), GadA (glutamate, tyrosine decarboxylase, or related PLP-dependent decarboxylase conserved domain family) superfamilies. In addition to those in *A. thaliana*, *L. brevis*, *S. cerevisiae*, and *T. reesei*, the number and distribution of domains in other species were highly consistent. In addition, the conserved motifs and domains of the two identified GAD proteins were very similar; both contained GAD domains, and the motif quantity distribution was consistent, showing high homology (Figure 7C). Combined with the molecular docking results, it can be seen that Ff-GAD2 has lower binding energy and stronger hydrogen bonding ability than Ff-GAD1, which makes the expression more stable. Subsequently, the Ff-GAD2 protein-coding gene was selected for functional verification.

Because the GABA content in *H. marmoreus* was significantly lower than that in *F. filiformis* and a mature genetic transformation system had been constructed in the early stage, *H. marmoreus* was selected as the heterologous host for the in vitro functional analysis of *Ff-gad2*. The similarity between *F4-Hm-gad2* and *Ff-gad2* was 55.70%, and *F4-Hm-gad2* had 13 exons and 12 introns (Appendix A). Bioinformatic analysis revealed that both F4-Hm-GAD2 and Ff-GAD2 have a conserved region, GadA, that belongs to the AAT-I superfamily (Appendix A). Multiple alignments of the GAD sequences from the two species suggested that GSTYTG (Motif-1) contained in both proteins might be a substrate binding site, and HVDAASGG (Motif-2) might be a highly conserved region of pyridoxal phosphate-dependent decarboxylase. The motif NASGHKYGMAYVGVGWVIWR may be a PLP-binding domain (Appendix A). Moreover, in addition to gene sequences, F4-Hm-GAD2 and Ff-GAD2 differed in spatial structure. In terms of secondary structure, F4-Hm-GAD2 had fewer α-helices (178/36.33%) but more irregular coils (244/49.80%) than Ff-GAD2 (Appendix A). In the tertiary structure, the prediction models of F4-Hm-GAD2 and Ff-GAD2 differed considerably (Appendix A). These differences in spatial structure might be responsible for the difference in the catalytic function of the two proteins, which resulted in the lower GABA content in *H. marmoreus*.

### 3.7. Functional Validation of GABA Synthesis Gene Ff-gad2 in F. filiformis

To verify the function of *Ff-gad2*, heterogeneously expressed Ff-GAD2 in the mycelium of the monokaryon of *H. marmoreus* was examined to determine its effect on GABA metabolism. The constructed plasmids pWY601, pWY603, and p018-empty vector were transferred into *A. tumefaciens* strain EHA105 for genetic transformation experiments. The respective putative transformations were randomly selected for DNA extraction, and the *Ff-gad2-gDNA*, *Ff-gad2-cDNA*, and *Pesdi1* gene fragments were amplified. The fragment sizes of the PCR products obtained were 2500, 1000–2500, and 750 bp, respectively, which were consistent with the expected 2518, 1698, and 765 bp, indicating that the exogenous DNA in the transformed plasmid was successfully transferred into the monokaryon genome of *H. marmoreus* (Figure 8A). The T-DNA insertion sites of the transformants of pWY601 and pWY603 were determined by re-sequencing. It was found that the transformants of pWY601 and pWY603 were all single-copy insertions. Except for the reverse insertion of pWY601-1, the other transformants exhibited forward insertions, distributed across five different genomic scaffolds, and the length of the plasmid fragments integrated into the scaffolds ranged from 4078 bp to 7305 bp. Despite these variations, both the *Ff-gad2* and *Pesdi1* genes were fully integrated into the genome (Appendix A). Quantitative real-time PCR was used to detect the expression of *Ff-gad2* in pWY601 and pWY603 transformants, compared with the expression of pWY603-3 transformants. The average relative expression in pWY601 transformations of 32.63 ± 5.70 times and in pWY603 transformations of 4.10 ± 3.43 times. The results showed that the expression of *Ff-gad2* in pWY601 was 7.96 to 1.39 times greater than that in pWY603, indicating that the intron of *Ff-gad2* contributed to the enhancement of gene expression (Figure 8B). An amino acid analyzer was used to determine the GABA content of the inverters. The GABA content of the transformed pWY601 reached 1973.18–3274.43 ng/mg, which was 283.90 ± 102.76% higher than that of the wild type. The GABA content of the pWY603 transformations reached 1104.67–1419.39 ng/mg, which was 85.40 ± 26.71% higher than that of wild type, indicating that the *Ff-gad2* gene can increase endogenous GABA content (Figure 8C). To explore the effect of transformation on mycelial growth, the growth rate and mycelial fresh weight were measured, and colony morphology was recorded. The results showed that the growth rate of transformations reached 2.52–2.69 mm/d, which was 9.39 ± 2.35% higher than that of the wild type, and the fresh weight of mycelium reached 0.49–0.62 g. Compared with the wild type, the weight increased by 18.44 ± 7.57% (Figure 8D), and there was no difference in colony morphology between the wild type and the transformations cultured at 23 °C for 14 d, indicating that the increase in endogenous GABA content increased the mycelial growth rate and mycelial biomass while having no notable effect on colony morphology (Figure 8E).

## 4. Discussion

*F. filiformis* is one of the most important edible fungi cultivated worldwide. It has significant economic and nutritional value and a long history of agricultural production. In previous studies, *F. filiformis* was misclassified because of its morphological characteristics, which are very similar to those of the European *Flammulina velutipes*. Subsequently, the researchers conducted a genomic analysis and identified 12 phylogenetic species [71]. Comprehensive studies have revealed differences between the East Asian *F. filiformis* and European *F. velutipes* [72,73]. *F. filiformis* cultivated in China, Japan, and Korea should be regarded as an independent species [74], which solves the problem of species attribution of *F. filiformis*. High-quality genomes not only provide new ideas for fungal taxonomic research but also a scientific basis for the precise acquisition and utilization of germplasm resources based on genome and genetic engineering in the future. Three-generation assembly and second-generation correction strategies were adopted, which have the characteristics of a high assembly rate and high base accuracy. A phylogenetic analysis of the 23 fungal species showed that *Fv-HL23-1* was closely related to *A*. *mellea*, *L*. *edodes*, and *S*. *commune*. In addition, 589 CAZymes were identified across the genome, indicating that *Fv-HL23-1* has a strong lignocellulose degradation capability and can utilize cellulose and hemicellulose as substrates for growth.

One of the most interesting studies of P450 in edible fungi is on the biosynthesis of *Ganoderma* triterpenes. Chen et al. [67] found that during the development of *G. lucidum*, the expression of 78 *CYP450* genes was positively correlated with the content change in *Ganoderma* triterpenes. Yang et al. [75] selected the high-expression gene *CYP512u6* to construct the expression vector, yielding recombinant strain *ZYN02*. The microplasmid was extracted, and an in vitro enzymatic reaction showed that it could promote the biosynthesis of *Ganoderma* triterpenes. Wang et al. [76] used *S. cerevisiae* to screen 219 *CYP450* genes in the genome of *G. lucidum* and found that the overexpression strain of *cyp5150l8* could produce *Ganoderma* triterpenes. Subsequently, Yuan et al. [77] found that *Ganoderma* acid production catalyzed by *CYP512W2* could be improved by two orders of magnitude. In this study, 108 CYP genes were identified in the genome of *Fv-HL23-1*, which could be divided into groups A–D, and FfCYPs on different branches may exercise different functions. There were 64 P450 genes with important functions annotated by KEGG analysis, among which the CYP51 family encodes key enzymes for the synthesis of ergosterol in *F. filiformis* [78], with strong specificity. Ergosterol is a kind of steroid substance used as a component of the fungal cytoplasmic membrane. It plays an important role in maintaining cell fluidity and integrity, maintaining cell osmotic pressure, regulating membrane protein function, and improving growth rate. In addition, Han et al. [79] used *Aspergillus oryzae* to characterize a P450 enzyme in *F. filiformis* and found that co-localization with *AxeB* resulted in the production of a novel compound, 3-oxo-axenol. The extensive presence and diversity of the CYP450 family in the genome of *F. filiformis* provide a rich gene pool for targeted screening in genetic and metabolic engineering to enhance specific metabolites or biological characteristics.

The GABA synthesis pathway includes the GABA shunt and the polyamine degradation pathway. The *gad* gene, encoding the key enzyme GAD in the GABA shunt, is closely related to GABA content. GAD is a pyridoxal phosphate-dependent enzyme that catalyzes the α-decarboxylation of L-Glu to GABA. *gad* genes in edible fungi are known to play important roles in GABA biosynthesis and physiological regulation; however, the structure–activity relationship of *gad* genes has not been thoroughly studied. In the present study, two Ff-GAD proteins, Ff-GAD1 and Ff-GAD2, were identified in the *F. filiformis* monokaryon *Fv-HL23-1* by homology alignment. Bioinformatic analysis showed that the evolutionary relationships of these two proteins were very close, and their secondary and tertiary structures and physicochemical properties were similar. Molecular docking simulations showed that the two proteins interacted with the natural substrate L-Glu and cofactor PLP to form a stable complex through hydrogen bonding. Among these, the residues Thr137, Ala138, Ser332, Ser334, and Asn336 of Ff-GAD2 formed hydrogen bonds with the aldehyde group of PLP and produced the lowest binding energies. These amino acid residues help to stabilize the phosphate group and pyridine ring of the PLP cofactor, the substrate in the unsaturated cavity of the binding pocket [80], and the expression of Ff-GAD2 protein; therefore, *Ff-gad2* was selected for the functional study. L-Glu mostly forms hydrogen bonds with carboxyl groups on protein residues when entering the active pocket of the Ff-GAD catalytic reaction, and the pyridoxal phosphate-dependent enzyme GAD catalyzes L-Glu irreversibly to GABA, mainly through the following steps: In the absence of substrate L-Glu, PLP covalently binds to the C-NHz of the Lys residue in the GAD active pocket to form an internal aldiimine structure. In the presence of L-Glu, PLP reacts with α-NHz to form external aldiimine, which is activated by decarboxylation reaction to form quinone intermediates. Finally, GABA reacts quickly with PLP to form an aldiimine, which releases GABA [81]. Whether the Ff-GAD proteins in this study have a similar reaction mechanism needs to be further examined with protein crystal structure analysis and three-dimensional structure analysis to determine the substrate specificity of the two Ff-GAD proteins.

Studies comparing *gad* derived from different species have found that enhancing or decreasing the expression of the gene encoding GAD can change the GABA content. Takayama et al. [16] overexpressed *SlGAD3*, the key GAD-coding gene in tomato, which increased the mRNA level in mature green and red fruits by more than 20- and 200-fold, respectively, and the GABA content was significantly increased by 2.7–5.2-fold compared with the wild type, and no abnormality was found in the development of fruits and organs. Lan et al. [82] identified six GAD proteins in the *Dimocarpus longan* Lour whole genome. The overexpression of *DlGAD5* enhanced the transcription and translation of the GAD-encoding gene and increased the activity of the GAD enzyme to 1.16 times that of the control group. Akter et al. [83] used CRISPR/Cas9-mediated gene editing in rice seedlings to trim the coding region of CaMBD in *OsGAD4*, resulting in a GABA content approximately nine times higher than that of the wild type, up to 11.26 mg/100 g DW. At the same time, this enhanced rice tolerance to abiotic stress. Additionally, Xie et al. [84] overexpressed *PagGAD2*, a key enzyme gene for GABA synthesis, to increase endogenous GABA content in poplar and found that GABA had a negative regulatory effect on the formation and growth of adventitious roots. With the development of molecular biology technology, *gad* genes from tomato, *D. longan*, rice, *Populus*, *L. brevis*, *A. thaliana*, and other organisms have been cloned and expressed, and the biochemical characteristics of the corresponding GAD proteins have been explained. However, the function of *gad* genes from edible and medicinal fungi in GABA biosynthesis has not yet been verified in vitro. In this study, based on the efficient genetic transformation technology system of *H. marmoreus*, we heterogenetically expressed the GAD2 gene *Ff-gad2*, which was unearthed from the genome of *Fv-HL23-1*. Compared with that in wild-type *H. marmoreus*, the GABA content of the transformations increased by 85.40–283.90%. The growth rate of the mycelium was increased by 9.39 ± 2.35%, and the fresh weight was increased by 18.44 ± 7.57%, suggesting that *Ff-gad2* could help to increase GABA content, mycelium growth rate, and biomass. In addition, the expression of *Ff-gad2-gDNA* was 7.96 ± 1.39 times higher than that of *Ff-gad2-cDNA*, suggesting that the full-length *Ff-gad2* gene of *Fv-HL23-1* was more conducive to the expression of GAD protein in *H. marmoreus*. In many eukaryotes, introns have been demonstrated to promote gene expression, and the main mechanism may be by increasing the transcription level and improving the efficiency of mRNA translation through the modulation of transcription speed, nuclear export, and transcriptional stability [85].

In the future, RNA interference and CRISPR/Cas9 can be used to knock out or knock down the *Ff-gad2* gene to further elucidate the function of the gene encoding the key enzyme in the GABA metabolic pathway and to determine the substrate specificity of Ff-GAD combined with the protein crystal structure. This study provides a theoretical basis for understanding the molecular mechanism of GABA synthesis in *F. filiformis* and other edible fungi, as well as for the targeted breeding of GABA-enhanced varieties based on metabolic pathways.

## Figures and Tables

**Figure 1 jof-10-00862-f001:**
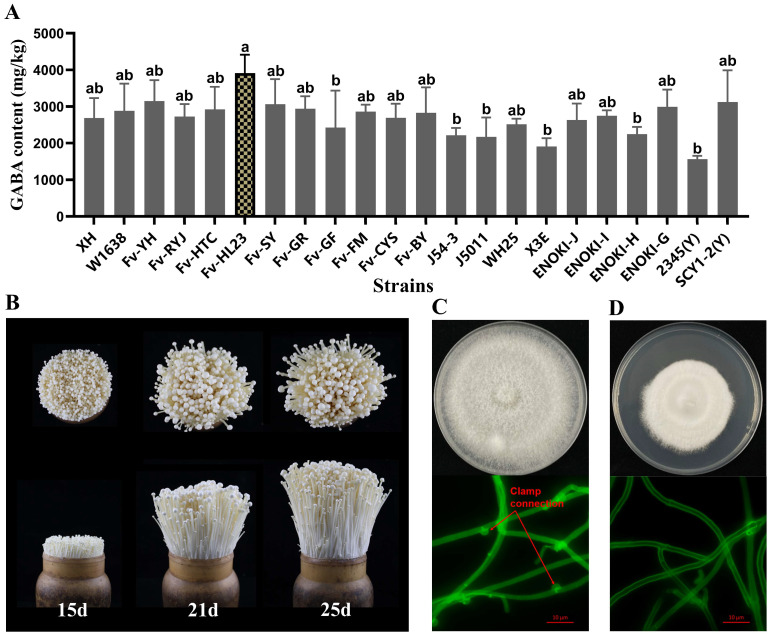
Screening, monokaryon isolation, and microscopic observation of *F. filiformis* strains with high GABA content: (**A**) Determination of GABA content in the fruiting body of *F. filiformis*. (**B**) Different growth periods in the fruiting body of the factory-cultivated *Fv-HL23* strain. (**C**) Growth plate and microscopic examination of *Fv-HL23* dikaryon hyphae. (**D**) Growth plate and microscopic examination of *Fv-HL23-1* monocytic hyphae. Different lowercase letters in (**A**) indicate a significant difference at *p* < 0.05. The red arrow in (**C**) indicates the clamping connection.

**Figure 2 jof-10-00862-f002:**
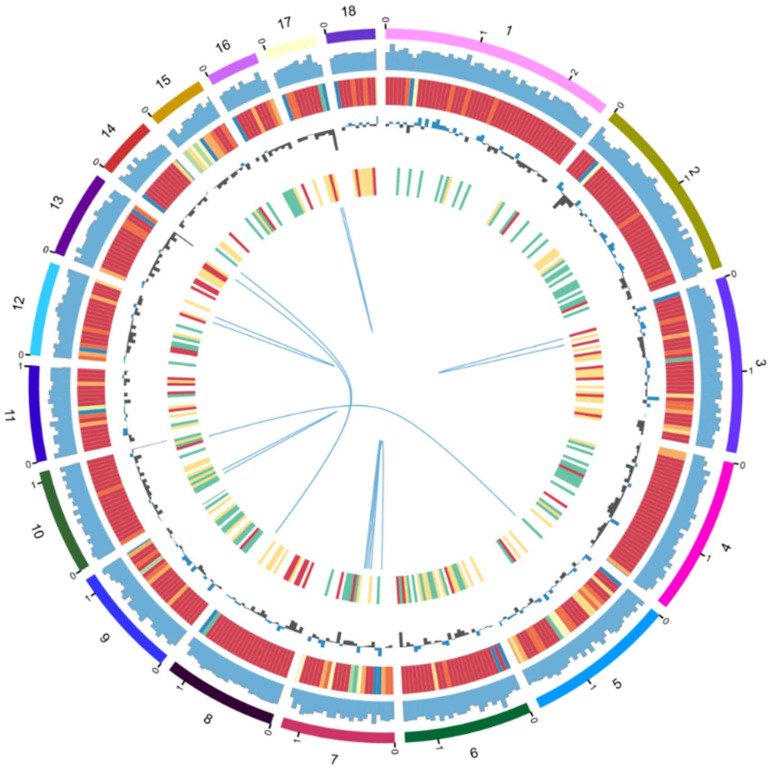
Overview of the *Fv-HL23-1* genome assembly. The circle diagram from outside to inside represents different chromosome lengths, gene density, repeat sequence density (red to blue, representing the increase in sequence density), GC content, number of CAZyme genes in the window (white, green, yellow, and red, representing the increase in the number of genes), and collinear block connection between chromosomes. The first 18 scaffolds of the genome were taken for mapping using Circos v0.69-3 software with a window size of 50 kb.

**Figure 3 jof-10-00862-f003:**
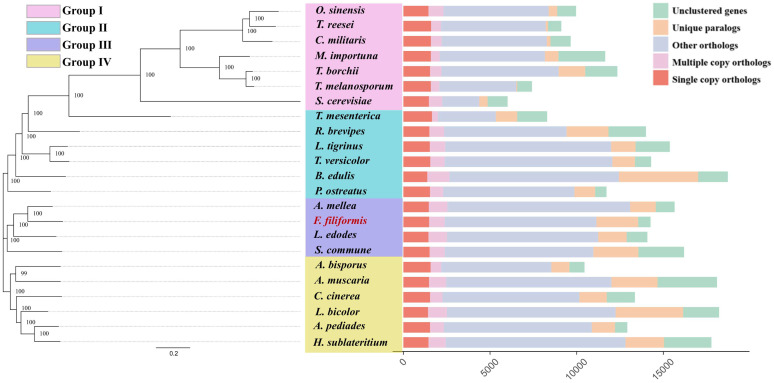
Evolutionary relationship analysis of *Fv-HL23-1* and 22 other fungal species. The analysis was performed based on single-copy orthologous genes. Tree scale = 0.2.

**Figure 4 jof-10-00862-f004:**
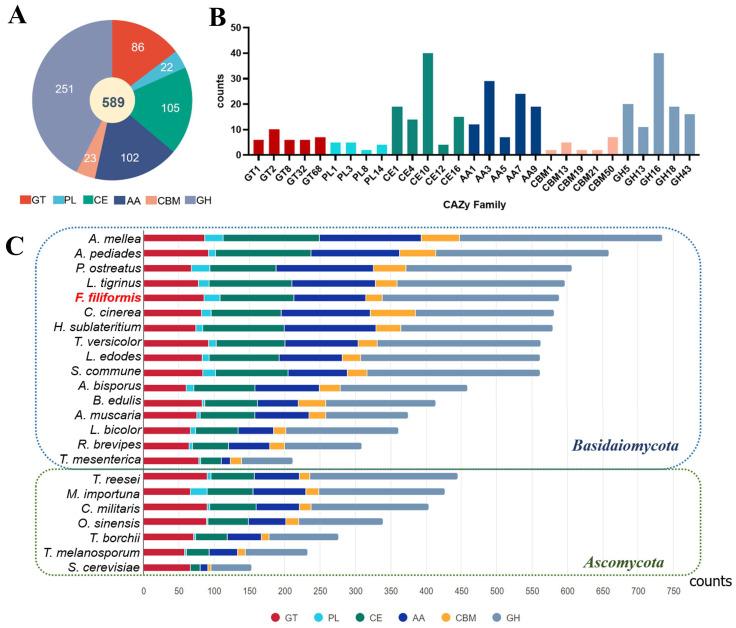
CAZymes in *Fv-HL23-1* and 22 other fungi: (**A**) Distribution of CAZyme categories in *Fv-HL23-1*. (**B**) Main gene numbers of CAZyme families in *Fv-HL23-1*. (**C**) Distribution of CAZymes in 22 other fungi. The strain names of these fungi are the same as those in Appendix A. Strain *Fv-HL23-1* is shown in (**C**) in red.

**Figure 5 jof-10-00862-f005:**
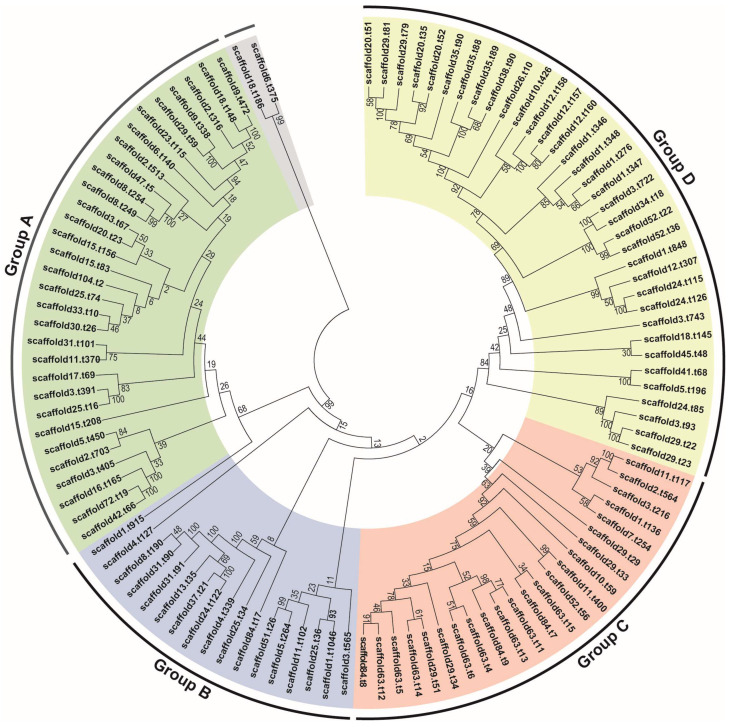
Phylogenetic evolutionary relationships of CYP450 proteins in *Fv-HL23-1*.

**Figure 6 jof-10-00862-f006:**
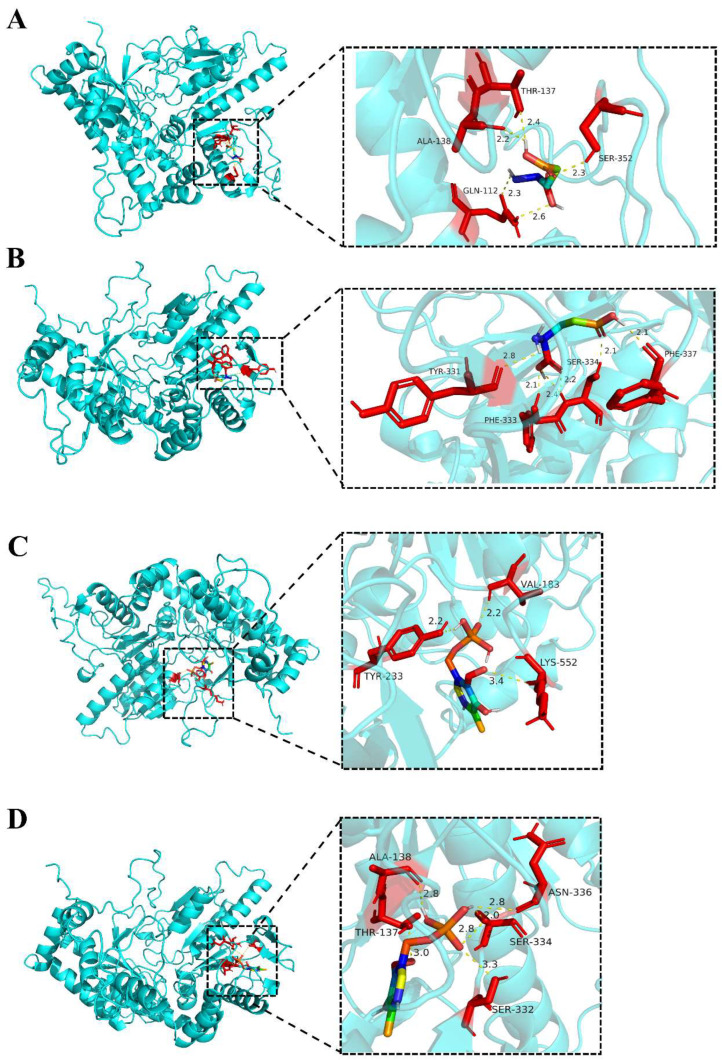
Molecular docking results for Ff-GAD protein with L-Glu and PLP: (**A**) Molecular docking results of Ff-GAD1 and L-Glu; (**B**) molecular docking results of Ff-GAD2 and L-Glu; (**C**) molecular docking results of Ff-GAD1 and PLP; (**D**) molecular docking results of Ff-GAD2 and PLP.

**Figure 7 jof-10-00862-f007:**
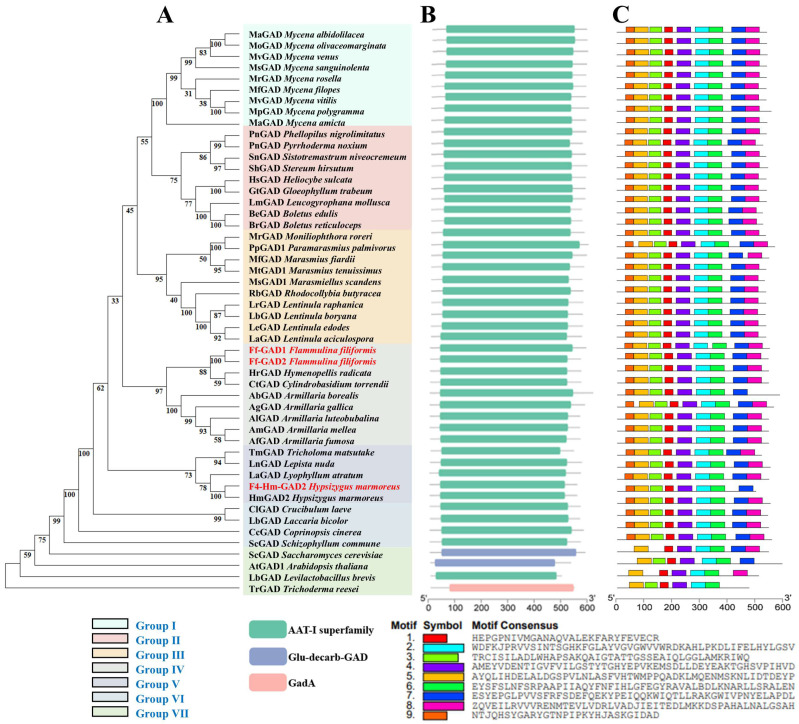
Phylogenetic analysis (**A**), conserved motifs (**B**), and domain analysis (**C**) of the Ff-GAD protein from *Fv-HL23-1* and GAD proteins from other sources. The GAD protein sequences were downloaded from the NCBI online database: KAF8136268.1 (*Boletus edulis*), KAG6376064.1 (*Boletus reticuloceps*), KAH7930877.1 (*Leucogyrophana mollusca*), XP_007863065.1 (*Gloeophyllum trabeum*), TFK55286.1 (*Heliocybe sulcata*), KAH8108454.1 (*Phellopilus nigrolimitatus*), PAV24364.1 (*Pyrrhoderma noxium*), XP_007301409.1 (*Stereum hirsutum*), KZS96066.1 (*Sistotremastrum niveocremeum*), KAJ7063082.1 (*Mycena amicta*), KAJ7684948.1 (*Mycena polygramma*), KAJ6513906.1 (*Mycena vitilis*), KAJ7172732.1 (*Mycena filopes*), KAJ7694132.1 (*Mycena rosella*), KAJ6455821.1 (*Mycena sanguinolenta*), KAF7338502.1 (*Mycena venus*), KAJ7349472.1 (*Mycena albidolilacea*), KAJ7905262.1 (*Mycena olivaceomarginata*), ESK85893.1 (*Moniliophthora roreri*), KAK7043692.1 (*Paramarasmius palmivorus*), KAJ8078534.1 (*Marasmius tenuissimus*), KAF9264894.1 (*Marasmius fiardii*), KAK7466895.1 (*Marasmiellus scandens*), KAF9065262.1 (*Rhodocollybia butyracea*), KAJ3773870.1 (*Lentinula raphanica*), KAJ4001425.1 (*Lentinula boryana*), KAJ4488196.1 (*Lentinula aciculospora*), KAJ3921114.1 (*Lentinula edodes*), RDB24023.1 (*Hypsizygus marmoreus*), KAF8061476.1 (*Lyophyllum atratum*), KAF8226168.1 (*Tricholoma matsutake*), KAF9466420.1 (*Lepista nuda*), KAF9050759.1 (*Hymenopellis radicata*), KIY69590.1 (*Cylindrobasidium torrendii*), KAK0450620.1 (*Armillaria borealis*), PBK96596.1 (*Armillaria gallica*), KAK0503002.1 (*Armillaria luteobubalina*), KAK0228300.1 (*Armillaria fumosa*), KAK0197355.1 (*Armillaria mellea*), XP_001878222.1 (*Laccaria bicolor*), TFK42928.1 (*Crucibulum laeve*), KAG2020630.1 (*Coprinopsis cinerea*), XP_050198490.1 (*Schizophyllum commune*), EDN64189.1 (*Saccharomyces cerevisiae),* AAA93132.1 (*Arabidopsis thaliana*), BAN07465.1 (*Levilactobacillus brevis*), XP_006965160.1 (*Trichoderma reesei*).

**Figure 8 jof-10-00862-f008:**
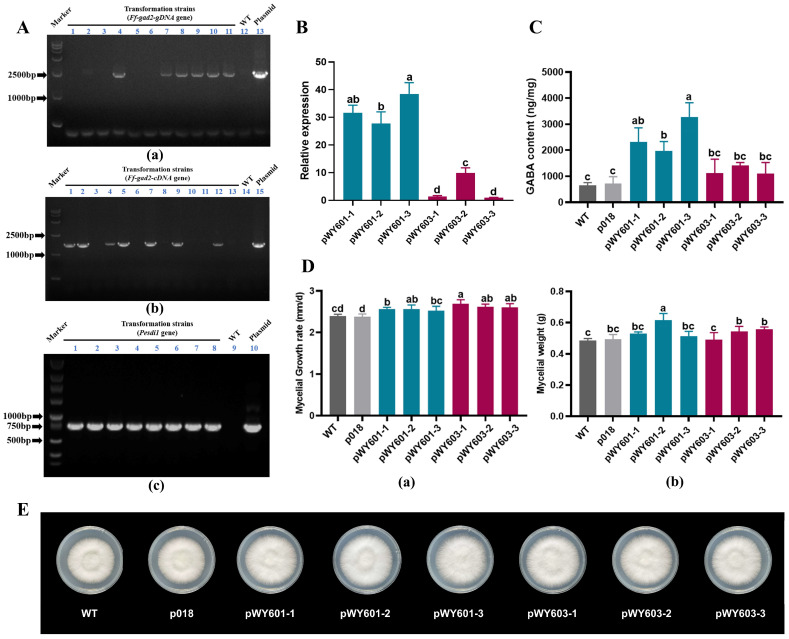
Acquisition and functional validation of the *Ff-gad2* transformations: (**A**) Amplification gel plots of the target fragments of the corresponding *Ff-gad2-gDNA* (a), *Ff-gad2-cDNA* (b), and *Pesdi1* (c) of the transformations. (**B**) Determination of the relative expression of *Ff-gad2* in pWY601 and pWY603 transformations. (**C**) Determination of GABA content in pWY601 and pWY603 transformations. (**D**) Mycelial growth rate (a) and mycelial weight (b) determination in *Ff-gad2* transformations. (**E**) Observation of mycelial morphology in eukaryotic transformations transfected with *Ff-gad2* gene. Different lowercase letters indicate a significant difference at *p* < 0.05.

**Table 1 jof-10-00862-t001:** Summary of source information of *F. filiformis* strains used in this experiment.

Strain Name	Origin	Cap/Stipe Colour	Source Information
*XH*, *W1638*, *Fv-YH*, *Fv-RYJ*, *Fv-HTC*, *Fv-HL23*, *Fv-SY*, *Fv-GR*, *Fv-GF*, *Fv-FM*, *Fv-CYS*, *Fv-BY*	Shanghai, China	Snow white/Snow white	Industrial cultivation
*J54-3*, *J5011*, *WH25*	Shanghai, China	Snow white/Snow white	SAAS-self-selection breeding
*X3E*	Shanghai, China	Snow white/Snow white	Enterprise-self-selection breeding
*ENOKI-J*, *ENOKI-I*, *ENOKI-H*, *ENOKI-G*	Malaysia	Snow white/Snow white	Industrial cultivation
*2345(Y)*	Shanghai, China	Light yellow/Snow white	Conventional cultivation
*SCY1-2(Y)*	Shanghai, China	Solid yellow/Solid yellow	Industrial cultivation

SAAS: Shanghai Academy of Agricultural Sciences.

**Table 2 jof-10-00862-t002:** List of primers used for gene manipulation and RT-qPCR analysis in this study.

Primers	Sequences (5′-3′)
Ff-gad2-gDNA-F	CTTCTGACTGACTTGAGGTAAATAGGTTAACATGCTCTCCAAGGTGACGAC
Ff-gad2-gDNA-R1	TCTAGACTACTTGTCATCGTCGTCCTTGTAATCACACGGCTTCGCATATGT
Ff-gad2-gDNA-R2	CTTCACTTCAAGTGCACACAACATATTCTAGACTACTTGTCATCGTCGTC
Ff-gad2-cDNA-F	CTTCTGACTGACTTGAGGT
Ff-gad2-cDNA-R	CTTCACTTCAAGTGCACACAA
Pesdi1-F	CTCATCTGGAAGGTGGCAGG
Pesdi1-R	TGGAGCGACGAGGATACAAC
q-Ff-gad2-F	TTTGAGCTGCACTACCTGGG
q-Ff-gad2-R	GTTCAAAGCGATTTGCCGGT
q-ACT1-F	CCGAGCGGAAGTACTCTGTG
q-ACT1-R	ATGCTATCTTGCCTCCAGCC

Note: GTTAAC is the *Hpa* I restriction enzyme cut site, and TCTAGA is the *Xba* I restriction enzyme cut site.

**Table 3 jof-10-00862-t003:** Summarized genome information for *Fv-HL23-1*.

Assembly Statistics	Scaffolds
Genome size (Mb)	40.96
Total sequence number	140
Total sequenced length (bp)	35,904,424
Maximum sequence length (bp)	2,467,745
Minimum sequence length (bp)	5510
GC (%)	49.62
N20 length (bp)	1,658,848
N50 length (bp)	917,125
N90 length (bp)	141,141
Total gene length (bp)	23,522,018
Gene Percentage of Genome (%)	65.51
Total number of genes	14,256
Average gene length (bp)	1649.9
Total exon length (bp)	19,949,251
Exon percentage of genome (%)	55.56
Average Exon Length (bp)	250.7
Total coding sequence (CDS) length (bp)	19,949,251
Average CDS length (bp)	1399.3
CDS percentage of genome (%)	55.56
Average intron length (bp)	54.7
Sequencing platform	PacBio CLR, Illumina

**Table 4 jof-10-00862-t004:** Repeat element analysis in the *Fv-HL23-1* genome.

Repeat Elements	Copies (Numbers)	Repeat Size (bp)	Percentage of the Assembled Genome
LTR elements	1164	375,774	1.05%
LINEs	276	28,892	0.08%
SINEs	6	316	0.00%
DNA transposons	563	49,884	0.14%
Simple repeats	58	6577	0.02%
Low complexity	4	467	0.00%
Satellites	38	4626	0.01%
Unclassified	104	22,885	0.06%
Total	2213	487,753	1.36%

Note: LTR, long terminal repeat; LINEs, long interspersed nuclear elements; SINEs, short interspersed elements.

**Table 5 jof-10-00862-t005:** List of protein subfamilies encoded by FfCYPs annotated using the KEGG database.

Name	Subfamily	Name	Subfamily
scaffold29.t59	CYP1A1	scaffold18.t145	CYP4F2_3
scaffold8.t249	CYP1A2	scaffold12.t307	CYP4F22
scaffold2.t316	CYP1B1	scaffold29.t79	CYP4Z
scaffold3.t391	CYP1B1	scaffold35.t90	CYP6
scaffold17.t69	CYP2AA_D	scaffold5.t196	CYP6
scaffold23.t115	CYP2D	scaffold1.t1046	CYP7A1
scaffold16.t165	CYP2E1	scaffold25.t36	CYP8B1
scaffold42.t66	CYP2E1	scaffold11.t117	CYP12
scaffold72.t19	CYP2E1	scaffold3.t216	CYP12
scaffold18.t148	CYP2J	scaffold25.t16	CYP17A
scaffold104.t2	CYP2R1	scaffold25.t74	CYP17A
scaffold30.t26	CYP2R1	scaffold3.t405	CYP17A
scaffold33.t10	CYP2R1	scaffold47.t5	CYP21A
scaffold11.t370	CYP2U1	scaffold2.t564	CYP27A1
scaffold6.t140	CYP2U1	scaffold29.t22	CYP28
scaffold13.t35	CYP3A	scaffold24.t126	CYP46A1
scaffold3.t743	CYP3A	scaffold7.t254	CYP49A
scaffold52.t36	CYP3A	scaffold3.t565	CYP51
scaffold29.t51	CYP3A	scaffold4.t127	CYP61A
scaffold24.t85	CYP3A4	scaffold1.t136	CYP67
scaffold24.t122	CYP4	scaffold20.t23	CYP78A
scaffold1.t848	CYP4A	scaffold9.t338	CYP81F
scaffold29.t81	CYP4B1	scaffold1.t915	CYP82G1
scaffold45.t48	CYP4B1	scaffold37.t21	CYP86A4S
scaffold10.t426	CYP4F	scaffold8.t190	CYP94C1
scaffold20.t35	CYP4F	scaffold2.t703	CYP98A9
scaffold20.t51	CYP4F	scaffold31.t101	CYP98A9
scaffold20.t52	CYP4F	scaffold12.t157	CYP313
scaffold3.t722	CYP4F	scaffold51.t26	CYP708A2
scaffold34.t18	CYP4F	scaffold41.t68	CYP735A
scaffold38.t90	CYP4F	scaffold18.t186	CYPD
scaffold52.t22	CYP4F	scaffold6.t375	CYPH

**Table 6 jof-10-00862-t006:** Predicted physicochemical properties of Ff-GAD proteins.

Name	Number of Amino Acids	Formula	Molecular Weight (kDa)	Theoretical pI	Instability Index	Aliphatic Index	Grand Average of Hydropathicity	Subcellular Localization
Ff-GAD1	554	C_2786_H_4310_N_750_O_812_S_18_	61.89	6.32	37.17	88.07	−0.238	mitochondrion
Ff-GAD2	536	C_2687_H_4173_N_725_O_782_S_14_	59.60	6.27	36.60	91.23	−0.216	mitochondrion

## Data Availability

Data are contained within the article. The public genomic information of *FV-HL23-1* is available in the Genbank database (JBGQRC000000000). The nucleotide sequence of *Ff-gad2-cDNA* was provided with GenBank ac-cession number PQ645160. The nucleotide sequence of *F4-Hm-gad2* was provided with GenBank accession number PQ586973.

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
