# Peer review of "Whole-Genome Sequence Analysis of Flammulina filiformis and Functional Validation of Gad, a Key Gene for γ-Aminobutyric Acid Synthesis"

_jof, 2024, doi:10.3390/jof10120862_

Round 1

Reviewer 1 Report

Major comment:

The submitted manuscript is devoted to the isolation and genome sequencing of Flammulina filiformis monokaryon. The study performed by the authors is well designed, properly described and comprehensively illustrated. The obtained genomes are of the good quality. Moreover, the authors performed an additional functional study of the genes – Ff-gad1 and Ff-gad2 – responsible for the synthesis of γ-aminobutyric acid. Heterologous expression of Ff-gad2 in the mycelia of Hypsizigus marmoreus makes the scope of the submitted article far exceed the format of a genome report alone.

Although the content of the manuscript is excellent, the English grammar and style need to be improved. Below I begin to write comments line by line, but I realized that I will have to correct almost every line of the manuscript.

English Language and Style:

Line 16 and Line 43: Please, double check – should it be “productive” or “produced”?

Lines 24-25 and Lines 90-92: Since you refer to all CAZymes, it is better to use “lignocellulose degradation ability” instead of “lignin degradation ability”.

Lines 31-36 and Lines 16-31 should be intertwined. Currently, it looks like two different abstracts – the detailed one and the short one – were artificially joined together. Additionally, explain abbreviations “Ff-GAD1” and “Ff-GAD2” upon the first introduction in the text, and clearly state that Lines 28-30 describe expression in the Hypsizigus marmoreus.

Lines 55-85: I subject to split this paragraph into two – (1) about previous genome findings; (2) about GABA.

Lines 91-94: This is awkward and ungrammatical sentence. Please, correct.

Lines 94-99: This is awkward and ungrammatical sentence. Please, correct.

Lines 112-113: “Complete yeast (CYM) liquid medium: …. and then sterilized” -> “Complete yeast (CYM) liquid medium was prepared as follows: …., and then sterilized”. Additionally, add the sterilization conditions. The other lines (Lines 116-123) also shod be corrected.

Please, correct the English in the whole Manuscript. It is annoying to write out here almost every third sentence.

In conclusion, I recommend major revision. Again, this is an excellent article, and it should definitely be published. Please, give it to someone with good command in English.

Please, see Major comments.

Reviewer 2 Report

1.Verification of gad2 Gene cDNA. The cDNA sequence of the gad2 gene, which is predicted based on Augustus software, requires further validation for accuracy in gene structure. Please provide Sanger sequencing validation of this gene's cDNA and upload the accurate sequence to a public database.

2.Plasmid Map and Primer Sequences. A schematic diagram of the plasmid should be provided, along with the primer sequences used to verify the exogenous fragments of Ff-gad2-gDNA and Ff-gad2-cDNA in the transformants. Additionally, indicate the locations of these amplification primers on the schematic diagram.

3.P450 Database for Gene Annotation. In the methods section, the P450 database was utilized for gene annotation. However, the results lack a description of the P450 gene family. Please supplement this information and provide the gtf file for gene prediction along with the gene annotation results corresponding to Section 2.4.3.

4.Expression Levels of gad in pWY601 and pWY603 Transformants (Lines 521-523). The authors report a significant increase in gad expression in pWY601 transformants compared to pWY603 transformants. The proposed explanation, "the intron of Ff-gad2 contributed to the enhancement of gene expression," is not supported by scientific rigor. Expression levels of different transformants are affected by factors such as plasmid copy number and genome insertion site, which the authors have not tested. Drawing conclusions solely based on differences in heterologous expression levels is insufficient. It is necessary to supplement the information with the location of the qRT-PCR primers on the gene sequence, as well as detection of plasmid copy number and genome insertion site.

5.Heterologous Transformation and Phenotype Observation. This study involves heterologous transformation, and phenotypically, all heterologous expression transformants exhibit very similar mycelium and GABA phenotype results. However, it should be noted that these phenotypes could potentially be attributed to the presence of the exogenous plasmid. Therefore, it is essential to include a control experiment with an empty vector.

6. Genetic transformation in Flammulina filiformis is well-established. Why was heterologous expression in Hypsizygus marmoreus chosen for gene function validation instead of directly in Flammulina filiformis?

7. What is the similarity between the gad gene in Hypsizygus marmoreus and that in Flammulina filiformis? It is recommended to supplement a conservation analysis of the gad gene and amino acid sequences between these two species.

8. The manuscript lacks tolerance tests for the strain Finc-W247-F4 against the resistance markers Carboxin and Cefotaxime. The authors directly provide screening concentrations, which is unreasonable. Please explain the rationale for selecting these two antibiotics for screening.

9. Can GABA be secreted outside the cell? Only the GABA content in the mycelium was detected, and not in the fermentation broth. Can this represent the true GABA production by the transformants?

10. Since the Ff-gad2 gene does not exist in wild-type Hypsizygus marmoreus, how was the relative expression fold of Ff-gad2 gene calculated in the transformants?

11. Multiple genome sequences of Flammulina filiformis have been published, and the analysis of the Flammulina filiformis genome is a primary content of this paper. It should be discussed in depth. Additionally, there are relevant literature reports on CAZyme genes and Cytochrome P450s in Flammulina filiformis. Please supplement these references and include discussions on them.

1.Source and Relationships of Dikaryon Strains. Please provide more detail information on the source of the "Dikaryon strains of F. filiformis" and elucidate the relationships among these strains. Line 102 to 104

2.Cultivation Techniques and Substrate Formulation. Detailed descriptions of the cultivation techniques and substrate formulation used for F. filiformis are required. Line 111

3.Sampling Criteria for Fresh F. filiformis Fruiting Bodies. Please supplement the criteria for sampling "fresh F. filiformis fruiting bodies," including their growth status and stage. Line 125

4.Biological Replication and Statistical Methods. The number of biological replicates conducted for all experiments and the statistical methods employed should be included. Section 2

5.Reference Species for Augustus v3.03 Gene Prediction. In Section 2.4.3, please specify the reference species used for gene prediction with Augustus v3.03. Line 149

6.Literature Source and Function of Pesdi1 Gene. Clarify the literature source of the "endogenous mutant gene Pesdi1 of Pleurotus eryngii," detail the function of this gene, and explain how it was applied in the screening of Hypsizygus marmoreus transformants. Line 199

7. Replace "the fruit of F. filiformis plants cultivated" with "The fruiting bodies of F. filiformis". Line 264

8. The clamp connections in Figure 1C are not clearly visible. Line 276

9. In Figure 7A, it is necessary to add the transformant numbers corresponding to each lane. Line 540

Round 2

Reviewer 1 Report

As I mentioned in my first review, the study performed by the authors is well designed, properly described and comprehensively illustrated. The obtained genomes are of the good quality. Moreover, the authors performed an additional functional study of the genes – Ff-gad1 and Ff-gad2 – responsible for the synthesis of γ-aminobutyric acid. Heterologous expression of Ff-gad2 in the mycelia of Hypsizigus marmoreus makes the scope of the submitted article far exceed the format of a genome report alone.

Previously, I was most concerned about grammar and style. Since English grammar and style have been greatly improved in the revised version, I have no further comments or suggestions for the authors.

This is an excellent article, and I recommend it for publication.

Please, see the Major comment

Reviewer 2 Report

I have no more comments.

I have no more comments.